# Chronicling Germany: An Annotated Historical Newspaper Dataset

**Christian Schultze**
High-Performance Computing
and Analytics (HPCA-Lab)
Universität Bonn

**Niklas Kerkfeld**
HPCA-Lab, Universität Bonn

**Kara Kuebart**
Institut für Geschichtswissenschaft
Universität Bonn

**Princilia Weber**
Institut für Geschichtswissenschaft
Universität Bonn

**Moritz Wolter**[*]
HPCA-Lab, Universität Bonn

**Felix Selgert**[*]
Institut für Geschichtswissenschaft
Universität Bonn

## Abstract

The correct detection of article layout in historical newspaper pages remains challenging but is important for Natural Language Processing (NLP) and machine learning applications in the field of digital history. Digital newspaper portals typically provide Optical Character Recognition (OCR) text, albeit of varying quality. Unfortunately, layout information is often missing, limiting this rich source's scope. Our dataset is designed to address this issue for historic German-language newspapers. The Chronicling Germany dataset contains 581 annotated historical newspaper pages from the time period between 1852 and 1924. Historic domain experts have spent more than 1,500 hours annotating the dataset. The paper presents a processing pipeline and establishes baseline results on in- and out-of-domain test data using this pipeline. Both our dataset and the corresponding baseline code are freely available online. This work creates a starting point for future research in the field of digital history and historic German language newspaper processing. Furthermore, it provides the opportunity to study a low-resource task in computer vision.

## 1 Introduction

Newspapers are essential sources of information, not just for modern readers, but particularly in the past when other communication channels like the internet or radio were not yet available. Even more importantly, newspapers allow researchers to study social groups' opinions and cultural values both now and then. This paper presents the *Chronicling Germany*-dataset, consisting of 581 annotated high-resolution scanned newspaper pages from the period between 1852 and 1924.

With the emergence of digital newspaper portals, using historical newspapers has become easier in recent years[2]. These portals provide text via OCR but lack reliable layout information, which

---

[*]equal supervision

[2]For Germany, e.g., the *Deutsche Zeitungsportal* ( https://www.deutsche-digitale-bibliothek.de/ newspaper/) and *zeit.punkt NRW* (https://zeitpunkt.nrw/)

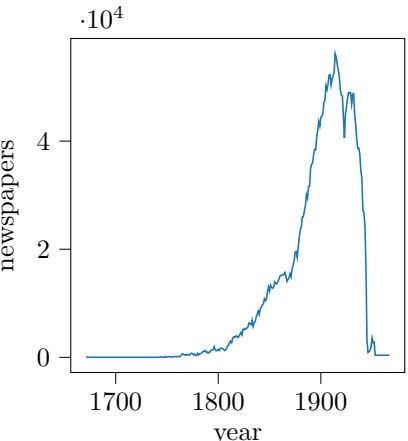 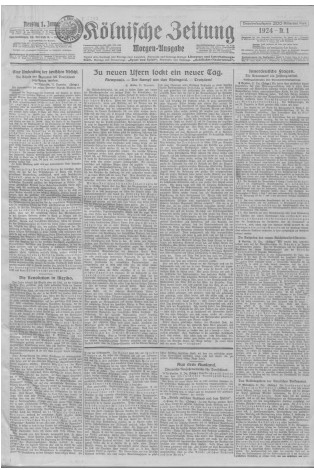

Figure 1: Left: Number of available digitized newspapers per year at `www.deutsche-digitale-bibliothek.de/newspaper` over time. Data from January 2024. Right: Front page of the *Kölnische Zeitung* from the $1^{st}$ of January 1924.

is essential for digital history applications, many of which would require newspaper articles to be treated as individual documents. Our dataset will help reduce the character error rate and considerably improve the detection of individual elements of a newspaper page, like articles or single advertisements. The former is important to prevent algorithms from connecting unrelated text regions and preserve the order in which text regions should be read. To this end, the text layout is systematically annotated using nine classes.

From a computer science view, a collection of successful approaches allows us to process modern documents (Blecher et al., 2023; Davis et al., 2022). For historic documents, large-scale data sets exist (Dell et al., 2024) but are mostly focused on English language material set in Antiqua-like typefaces. For continental European languages, existing datasets are much smaller (Abadie et al., 2022; Kodym and Hradis, 2021; Clausner et al., 2015; Nikolaidou et al., 2022).

Until more annotated data becomes available, the processing of historical continental European newspaper pages is, therefore, a low-resource task, highlighting the need for more data. While low-resource tasks are well-established in natural language processing (Adams et al., 2017; Fadaee et al., 2017; Hedderich et al., 2021; Zoph et al., 2016), low resource settings remain under-explored in computer vision (Zhang et al., 2024). Historical German newspapers are interesting in this context due to their dense layout (see also Supplementary Figure 6) as well as the Fraktur font. Fraktur differs significantly from the Antiqua typefaces that dominate modern Western texts. To the modern eye, Fraktur letters appear dense. Furthermore, in addition to the font, our dataset features the archaic 'long s' or 'ſ', which is no longer used today. The 'sz' or 'ß' is specific to the German language and also appears in the data. Historically, it emerged when the common combination 'ſz' merged into a single letter 'ß', unlike the 'long s' it still appears in contemporary texts. The aforementioned differences limit our ability to transfer existing solutions that were designed for modern documents or English-language historical newspapers. This motivates the collection of additional data.

The German newspaper processing task is also highly relevant to scholars of history. Especially in the 19th century, local communities, interest groups, and political parties created their own newspapers. The *Deutsche Zeitungsportal*[3] counts 698 German newspapers in 1780, this number rose to over 14,000 in 1860 and peaked at 50,848 papers in 1916 (see Figure 1). Plenty of digitized pages are available, which will allow researchers to systematically search for cultural values and historical change. Unfortunately, untrained modern human readers struggle with font differences, limiting the usefulness of unprocessed data to researchers lacking this specific skill. Thus, creating a pipeline capable of accurately processing this vast amount of data to a format readable to both a machine and a researcher without specific language and typeface skills is an important step in making these resources accessible.

---

[3]https://www.deutsche-digitale-bibliothek.de/newspaper/

Additionally, the layout of German historical newspapers is often complex, consisting of several columns, multiple horizontal sections and up to 500 elements to annotate per page. To create this dataset, eleven student assistants with a background in history have spent a total of 1,500 hours annotating the layout of our 581 pages. These include approx. 1,900 individually annotated advertisements, that consist of approx. 5,700 polygon regions. We also provide ground truth text annotations, which are not as costly since we start from network-generated OCR-output and correct errors. Overall, our dataset includes approx. 26,000 layout polygon regions as well as approx. 330,000 text lines.

Our dataset features sections and elements that are especially challenging for OCR and baseline models. For example, advertisement pages mix large and small font sizes and include drop capitals, where the initial letter of an advertisement spans over multiple rows but is read as part of the first row. Both features are a challenge for the baseline detection task. Other challenges are fractions in stock exchange news and abbreviations in lists of casualties.[4]

In summary, this paper makes the following contributions: (1) We introduce the *Chronicling Germany*-dataset consisting of 581 manually annotated high-resolution pages. (2) We establish a baseline recognition pipeline for the layout detection, text-line recognition, and OCR-tasks. (3) We verify generalization properties using 24 historic newspaper pages from the earlier 1785 - 1866 period. We observe good generalization performance.

The dataset and code for our pipeline are freely available online.[5]

## 2 Related Work

Very early text recognition systems worked with separately designed systems for line detection, baseline fitting, word detection, and word recognition (Smith, 2007). With the increased adoption of deep learning methodology in the field, neural networks took over many of these tasks until only text line detection and text detection remained as separate tasks (Zhang et al., 2016). The process culminated recently. Kim et al. (2022) propose to train transformer networks directly on images and annotated text without any intermediate steps. Their networks combine a swin transformer (Liu et al., 2021) with Bart decoder (Lewis et al., 2019). During an initial pre-training, their encoder is trained on two million synthetics and eleven million scanned documents. The decoder initially starts from weights pre-trained on multilingual text data. Full system training relies on 800 Latin alphabet receipts, 1,500 Chinese train ticket images, 20 thousand business cards, and 40 thousand Korean receipts. Using a similar system Blecher et al. (2023) trains an OCR engine to recover latex code from scans or PDFs of academic documents. Surprisingly, their network generalizes to old mathematical literature. However, Arxiv papers do not resemble historical newspapers. Therefore, transferring these results to the German historic newspaper domain remains challenging.

### 2.1 Historical Newspaper Processing

Unfortunately, from a digital history perspective, many modern systems focus on recent data and suffer from poor performance in a historical setting. The current situation has led to a large body of OCR error correction work (Carlson et al., 2023), highlighting the need for specialized data sets and software. Liebl and Burghard (2020), for example, combine existing open-source components for this task.

Related datasets include the Europeana corpus (Clausner et al., 2015). The dataset contains *528* annotated pages from European sources. More recently Dell et al. (2024), published perhaps the largest historical newspaper dataset to date. Their dataset also includes layout annotations. Our work complements these existing datasets by additionally providing compatible annotations for German historical newspapers that differ significantly from other Western European and American newspapers. Furthermore, we annotate advertisements in detail, which significantly add to the complexity of the OCR-task (Dell et al., 2024). Advertisements are particularly interesting to scholars of economic history who are interested in labor markets, for example.

---

[4]Our dataset includes pages from 1866, when the Austro-Prussian War was raging in the German Bund.

[5]Code: `https://github.com/Digital-History-Bonn/Chronicling-Germany-Code` Dataset: `https://gitlab.uni-bonn.de/digital-history/Chronicling-Germany-Dataset`

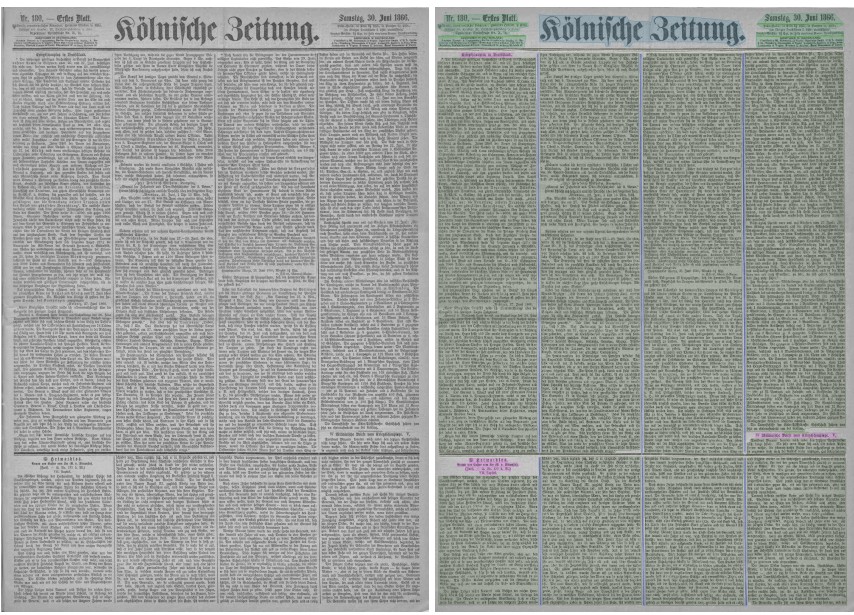

Figure 2: A *Chronicling Germany* page with its corresponding annotation side by side.

## 2.2 Common processing pipeline elements

**Layout Segmentation** is a longstanding task in document processing. For example, dhSegment (Oliveira et al., 2018) propose a UNet structure based on the popular ResNet50 architecture (He et al., 2016). As described by Ronneberger et al. (2015), the network features a contracting and an expanding part. The contracting subnetwork uses ResNet50 as an encoder, and an additional expansive subnetwork produces segmentation maps at the resolution of the original input. Transformer-based solutions trained on modern documents are available for similar tasks (Davis et al., 2022). However, Convolutional Neural Networks (CNNs) are cheaper to run (Dell et al., 2024) and require less training data, making them a budget-friendly solution.

**Baseline-detection** or text-line detection, means finding the straight line that connects the base points from each letter. Early work employed quadratic splines for this task (Smith, 2007). Modern solutions often employ architectures devised for segmentation or object detection tasks. Kodym and Hradis (2021) for example choose a U-Net, while Dell et al. (2024) work with YOLOv8.

**Optical Character Recognition (OCR)** is an important tool in digital history. Liebl and Burghard (2020), successfully work with a topological feature extraction step followed by a classifier as described by Smith (2007) for the digitization of the *Berliner Börsen Zeitung*. Following Breuel (2007), Kiessling (2022) uses a Recurrent Neural Network (RNN) based system. Dell et al. (2024) apply the contrastive learning approach presented by Carlson et al. (2023). Using a vision encoder, characters are projected into a metric space. The system works because patches containing the same character will cluster together.

## 3 The Chronicling Germany Dataset

Our Dataset contains pages from the *Kölnische Zeitung*, mostly from 1866, specifically from the period of the austro-prussian war. Of these 416 pages, 15 pages contain only advertisements with approx. 1,900 individual advertisement blocks. We also include 141 pages from 1924, as well as 24 special editions from 1852-1888. Polygons placed by our expert human annotators capture the layout for each page. All annotations are stored in PAGE-XML files. The Polygons capture different text-region types. Subclasses can exist within these. Each region type has a unique XML tag: `TextRegion`, `SeparatorRegion`, `TableRegion` and `GraphicRegion`. Graphic regions are always assigned the class `image`. Within text regions, we include the following classes: `paragraph`, `header`, `heading`, `caption`, `inverted_text`. Within table regions, the only possible subclass is `table`. To facilitate

Table 1: Label distribution-percentages per pixel in the dataset.

| label | class | frequency |
|-------|-------|-----------|
| 0 | background | 39.49% |
| 1 | caption | 0.74% |
| 2 | table | 2.90% |
| 3 | paragraph | 54.03% |
| 4 | heading | 0.94% |
| 5 | header | 0.68% |
| 6 | separator vertical | 0.62% |
| 7 | separator horizontal | 0.58% |
| 8 | image | 0.016% |
| 9 | inverted text | 0.014% |

correct reading order detection, we introduce the separator subclass `separator_vertical,` and `separator_horizontal`. Vertical separators highlight different columns of a page. Horizontal separators split the page into sections and are relevant for the reading order if they span over multiple columns. Otherwise, they are found at the beginning of a new article or between caption or header elements. The header category covers the newspaper's name, which appears at the top of the front pages. To the left and right of the newspaper name, historical newspapers often have smaller blocks with additional information, such as the name of the editor-in-chief, the publication date, or the subscription price. These polygons are annotated as captions. Polygons that cover paragraphs, headlines, and tables are annotated, respectively. The result of this annotation process is shown in Figure 2. Overall, the dataset includes 26,255 polygon regions.

We primarily use a combination of the classes described above to annotate the historic advertisements. We have decided not to introduce new classes to avoid confounding the model's training. This applies, in particular, to the separator classes. Therefore, we use the classes `separator_vertical` and `separator_horizontal` for the annotation of separator regions around individual advertisements. Advertisements tend to use text blocks with bigger fonts. To be consistent with our annotations, we mark these as `heading`. For the same reason, the normal-sized text is annotated as `paragraph`. Additionally, we include the classes `inverted_text` and graphic elements as `image`. These are present, especially in the advertisement pages, as well as the 1924 pages. Table 1 illustrates this numerically. The two classes `inverted_text` and `image` are only present in a subset of the data, which explains its low share of pixels overall.

Regions of each page have a reading order number assigned to them. These numbers are assigned automatically and not corrected manually. Reading order is not the main scope of this dataset. Automatic assignment leads to satisfactory results for most pages. For advertisement pages, however, it does not. Yet, advertisements don't need a meaningful reading order, as they are comprised of elements that are independent of each other.

In addition to the layout data, we include transcribed text divided into text lines. In our dataset, each text line is comprised of a polygon, which contains all characters, as well as a baseline and the corresponding text. Baselines and text transcriptions are initially generated automatically using the pipeline proposed by Kodym and Hradis (2021), and then corrected by expert annotators. Line polygons and baselines are only corrected when there are significant mistakes. This is especially the case within the advertisement pages, where some initial letters of advertisements span over more than one line. Correct drop capital detection is challenging for current text-line detectors. For the transcription, we concentrate on correcting lines with low confidence or containing many special characters. Overall, our dataset includes 330,281 text lines. The text correction process is ongoing. To date, we have corrected 124 pages. All pages will be ready in time for the Conference in December. The transcription follows the OCR-D guidelines, level 2 (Johannes Mangei, 2024). This means the text is transcribed in a visual style, preserving, for example, the archaic 'long s' or 'ſ'. For a complete discussion see supplemental section A.3.

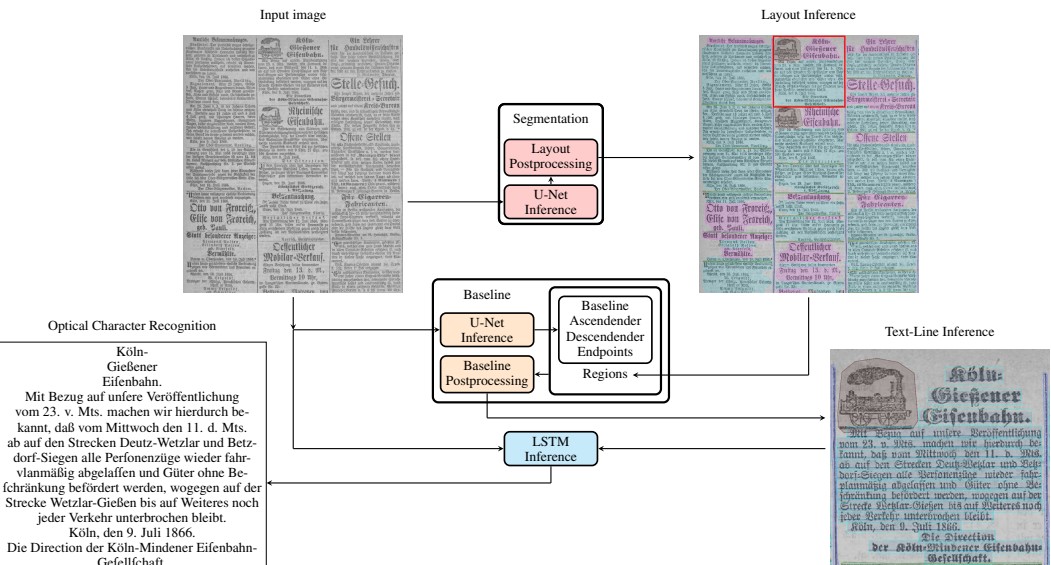

Figure 3: Flow chart of the entire prediction pipeline. The layout detection, text-line inference and Optical Character Recognition (OCR)-tasks use separate networks each. The output is machine-readable and can be processed further. For example in a machine translation step.

## 4 Experiments and results

**Data**: We work with fixed train, test and validation splits. We train on 492 pages with 30 validation pages and finally use 59 for testing.

**Pipeline**: Figure 3 gives an overview of our pipeline. Overall, we employ two U-Nets for Layout recognition and text-line detection and, finally, a Long Short-Term Memory (LSTM) cell for OCR. The pixel-wise layout inference is converted into polygons during the post-processing step. We use targets like Kodym and Hradis (2021) for training the baseline U-Net. The model recognizes baselines, ascender, descender, and endpoints, which are converted into line region and baseline polygons during post-processing. The post-processing code is an adapted version from Kodym and Hradis (2021). Contrary to their approach, we use the layout regions from the previous step to cut out parts of the image and identify all baselines for each region. These baselines are then used as input for the LSTM OCR model and the original image. The pipeline is sensitive to the character resolution. A small letter "a", for example, should be about 20x20 pixels in size. If the resolution deviates significantly (more than five pixel in either dimension), we rescale the input images accordingly.

### 4.1 Layout-Segmentation

**Training**: Our layout segmentation setup follows Oliveira et al. (2018). For layout training, all pages are scaled down by a factor of 0.5 and split into 512 by 512-pixel crops. Cropping leads to 34,376 training crops overall. During training, we work with 24 crops per batch per graphics card. The training runs on a node with four graphics processing units (GPUs). Consequently, the effective batch size is 96, with 358 training steps per epoch. Initially, optimization of the contracting network part can start from pre-trained ImageNet weights, while optimization of the expanding path has to start from scratch. The expanding subnetwork starts with the encoding from the contracting network and produces a segmentation output at the input resolution. To improve generalization, input crops are augmented using rotation, mirroring, gaussian blurring, and randomly erasing rectangular regions. An AdamW-Optimizer train this network with a learning rate of 0.0001, with a weight decay parameter of 0.001 for 50 Epochs in total, while using early stopping to save the best model. We explore transfer learning via pre-training on the Europeana-dataset (Clausner et al., 2015). In this case we initialize the encoder using ImagNet weights, train on Europeana first and continue training on our data. We compare to a network trained using ImageNet-weights only. In other words, we evaluate the effect of Europeana pre-training by working only with ImageNet pre-training.

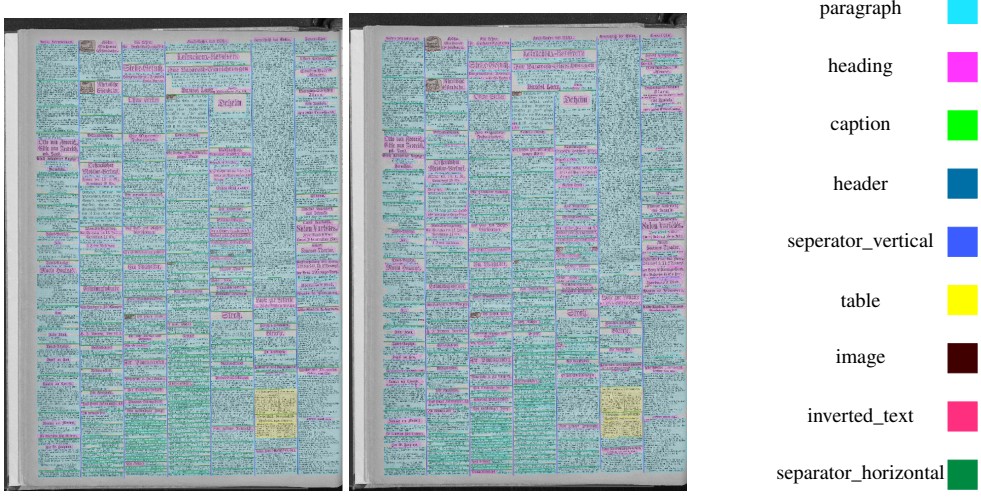

| paragraph | ![] (cyan) |
| heading | (magenta) |
| caption | (green) |
| header | (teal) |
| seperator_vertical | (blue) |
| table | (yellow) |
| image | (dark red) |
| inverted_text | (pink) |
| separator_horizontal | (dark green) |

Figure 4: Target labels on the left and segmentation prediction on the right. The top left part of this advertisement page also appears in Figure 3.

Table 2: Layout detection results. This table lists Intersection over Union (IoU) values for all individual classes, as well as overall.

| | Intersection over Union (IoU) [%] | |
| --- | --- | --- |
| class | ImagNet-Init | Europeana-Transfer |
| background | $0.89 \pm 0.001$ | $0.89 \pm 0.006$ |
| caption | $0.71 \pm 0.055$ | $0.75 \pm 0.024$ |
| table | $0.66 \pm 0.061$ | $0.64 \pm 0.055$ |
| paragraph | $0.97 \pm 0.001$ | $0.97 \pm 0.001$ |
| heading | $0.69 \pm 0.025$ | $0.68 \pm 0.015$ |
| header | $0.75 \pm 0.050$ | $0.75 \pm 0.027$ |
| separator vertical | $0.71 \pm 0.015$ | $0.73 \pm 0.013$ |
| separator horizontal | $0.66 \pm 0.055$ | $0.71 \pm 0.012$ |
| image | $0.36 \pm 0.050$ | $0.29 \pm 0.015$ |
| inverted text | $0.24 \pm 0.068$ | $0.15 \pm 0.054$ |

**Results**: Table 2 lists network performance. We compute Intersection over Union (IoU) values for individual classes. IoU is a widespread metric for segmentation tasks (Szeliski, 2022). Generally, we find good performance of the trained network, although the especially rare classes `image` and `inverted_text` are not recognized as well. Figure 4 presents an advertisement page from our test set with ground truth and prediction from the best pre-trained model side by side. Overall, Europeana pre-training improves separator recognition but does not help with images or inverted text, which are not annotated in the Europeana dataset.

## 4.2 Baseline Detection

**Training**: Following Kodym and Hradis (2021) we train an U-Net for the text-baseline prediction task. The raw input image as well as ground truth baselines serve as starting points for the optimization. The training process minimizes a joint text-line and text-block detection objective as introduced by Kodym and Hradis (2021).

We run an AdamW-optimizer with a learning rate of 0.0001 and a batch size of 16. During training, inputs are randomly cropped to 256 by 256 images. To improve the robustness of the resulting network the input pipeline includes color jitter, gaussian blur, random grayscale and gaussian blur perturbations during training.

Table 3: Baseline detection results. We measure performance in precision, recall and F1 score. Detected lines are matched with ground truth lines and are considered a true positive if the predicted line has an IoU score of more than 0.7 when compared with the corresponding ground truth line. Results are averaged over all test pages.

| Model | precision | recall | F1 score |
|-------|-----------|--------|----------|
| UNet | $0.910 \pm 0.008$ | $0.884 \pm 0.008$ | $0.896 \pm 0.007$ |

Table 4: Optical Character Recognition (OCR) results. Levenshtein distance per character appears in the first column. We computed the percentage of completely error-free lines for each model. The second column lists these results. Finally, we consider a line to have many errors if we observe a Levenshtein distance of more than 0.1 per character. We report the percentage of many error lines in the final column.

| Model | Levenshtein-Distance | completely correct [%] | many errors [%] |
|-------|----------------------|------------------------|------------------|
| UB Mannheim (2024) | 0.020 | 47.1 | 6.3 |
| transfer (ours) | $0.017 \pm 0.0009$ | $69.652 \pm 0.288$ | $5.075 \pm 0.216$ |
| random (ours) | $0.016 \pm 0.0013$ | $69.140 \pm 0.352$ | $5.146 \pm 0.318$ |

**Results**: We measure precision, recall, and F1 score (see Table 3). Generally, we observe values around 0.9. These observations are in line with Kodym and Hradis (2021), who observe similar numbers on the cBAD2019 dataset (Diem et al., 2017).

### 4.3 Optical Character Recognition (OCR)

**Training** Following Kiessling (2022) we train a LSTM-cell for the OCR-task. We employ baselines to extract individual text lines. Alongside the annotations, which have been checked by our human domain experts, these serve as input and ground truth pairs. Adam (Kingma and Ba, 2015) optimizes the network with a learning rate of 0.001. Optimization runs for a total of eight epochs with a batch size of 32 sequences. We used early stopping to prevent the model from overfitting. We include pixel-dropout, blur, rotation and see-through-like augmentations during training to improve generalization.

**Results**: We compare our results to a comparable pipeline developed by the Universitätsbibliothek Mannheim (Jan Kamlah, 2024) and observe improved results (see: Table 4), for an optimization starting from a randomly initialized RNN-cell. We also explore fine-tuning the model from Jan Kamlah (2024), which marginally improves the ratio of completely correct lines.

### 4.4 Overall pipeline performance

So far, we have evaluated components individually using ground truth inputs from previous steps. We additionally evaluate the complete pipeline on the test set. For each component, we choose the model with the best results and use the result of each component for the next one. Then, we evaluate the resulting transcription with our ground truth. All predicted and ground truth lines are matched based on the intersection over the minimum of the corresponding text lines. Lines without a match were paired with an empty string. Our pipeline achieves an overall Levenshtein distance per character of 0.0204 across the entire test set. Overall 97.96% of all output characters are correct.

## 5 Pipeline-Generalization

In this section, we report tests for the generalization properties of our pipeline.

**Test-Data**: In order to verify generalization, we work with annotated newspaper pages from different papers and a different period. Overall, 24 high-resolution pages from four newspapers from the period between 1785 and 1866 are annotated with manually corrected text. This out-of-domain test set contains approximately 250 regions and 2,400 test lines. Three of the four newspapers are set in Fraktur, and one is in Antiqua.

Figure 5: Generalization test set sample image. This figure shows a page element with detected baselines on the left. The right side presends the automatically created transcription.

**Inference**: We run the entire pipeline on this new generalization test set. Overall, we measure a Levenshtein distance per character of 0.0466, so 95.34% of characters are correct. Figure 5 presents an example taken from a 1785 issue of the Schwäbischen Merkur. The sample is a report from Portugal, which we deem entertaining from a modern perspective. Readers learn that hot-air balloons or "aeroſtatiſche Maſchinen" where banned "last year" because hot-air balloons are "incompatible with the omnipotence of god". Linguistically, the sample is close enough to modern German to be machine-translated.

# 6   Limitations and social impact

This dataset contains newspaper pages set in fraktur-letters. The font is very different from modern fonts. The 'long s' or 'ſ', for example, is completely foreign to modern eyes. While our generalization dataset also includes four pages in Antiqua font which have been predicted with sufficient accuracy, networks trained exclusively on our dataset are not likely to outperform more specialized networks on modern newspaper pages.

Ideally, our work will enable the processing of millions of pages of historical data, making vast resources easily available to future researchers who can then build upon the transcribed source material, for example, with machine translation and NLP pipelines. Countless research questions concerning economic, societal, political and scientific development can be addressed with such data. For a more detailed description of the relevance of such data for historical research, see Supplementary Section A.2. We hope this dataset will help to improve our understanding of the past. We therefore expect a positive impact on society as a whole.

# 7   Conclusion and future work

This work introduces the *Chronicling Germany*-dataset, a neural network-based processing baseline with test-set OCR-accuracy results. Our paper creates a starting point for researchers who wish to improve historical newspaper processing pipelines or are looking for a low-resource computer vision challenge. To create the dataset, history students spent 1,500 hours annotating the layout of our 581 pages. The dataset includes 1,900 individually annotated advertisements. Furthermore, we introduce an out-of-distribution test set of 24 pages. We verify baseline pipeline performance on these out-of-distribution pages. By following the OCR-D annotation guidelines (Johannes Mangei, 2024) we ensure our annotations' compatibility with concurrent and future work.

**Acknowledgments**

We thank the University of Bonn's transdisciplinary-research-areas TRA1 (Mathematics, Modelling, and Simulation of Complex Systems) and TRA4 (Individuals, Institutions, and Societies) for funding the data annotation. Furthermore, research was supported by the Bundesministerium für Bildung und Forschung (BMBF) via its "BNTrAInee" (16DHBK1022) and "WestAI" (01IS22094E) projects. The authors gratefully acknowledge the Gauss Centre for Supercomputing e.V. (www.gauss-centre.eu) for funding this project by providing computing time through the John von Neumann Institute for Computing (NIC) on the GCS Supercomputer JUWELS at Jülich Supercomputing Centre (JSC).

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

# A   Supplementary

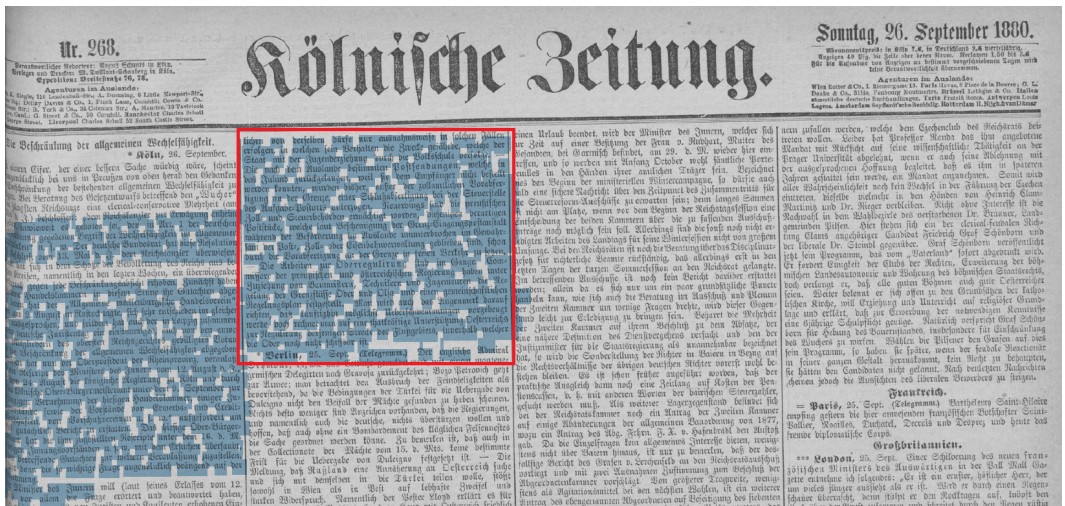

Figure 6: Layout recognition error in the *Kölnische Zeitung*. A researcher tried to select text in the first column on the very left but the column layout it not understood correctly. This page was published on September 26, 1880. Digital versions are available at the *zeit.punkt NRW* website. Layout recognition and transcription generated by Transkribus.

## A.1   Acronyms

**CNN**  Convolutional Neural Network

**GPU**  graphics processing unit

**IoU**  Intersection over Union

**LSTM**  Long Short-Term Memory

**NLP**  Natural Language Processing

**OCR**  Optical Character Recognition

**RNN**  Recurrent Neural Network

## A.2   Machine learning is important for the study of history

Figure 1 illustrates the breakthrough of the newspaper industry in the 19th century.[6] While the number of newspapers listed in the *Deutsche Zeitungsportal* grew at a rate of 2.9 percent p.a. in the first two-thirds of the 19th century, the increase rose to 3.4 percent p.a. after the foundation of the German Empire. There were three main reasons for this increase: firstly, the literacy of the population increased over the century. Secondly, considerable technological advances made it easier to produce a newspaper. Thirdly, state control of newspapers declined from the middle of the century.

---

[6]Note that the Deutsche Zeitungsportal does not collect all historical newspapers. There is probably a selection bias towards more prominent outlets with extended publication periods. However, on the whole, figure 1 should reflect the development of the newspaper market in Germany quite well.

A significant milestone was the Press Act of 1874, which finally abolished censorship (although some restrictions remained so that even after 1874, there was no complete freedom of the press in the German Empire). Nevertheless, no later than the last third of the 19th century, a mass market for print media had emerged in Germany, which was served by many newspapers whose content and political orientation were very heterogeneous.

Historical newspapers contain a wealth of information about past societies. They provide information about the spatial occurrence of events, about contemporary perceptions of social and economic change, and allow tracing of cultural change. Blevins (2014), for example, uses the mentioning of place names in the *Houston Post* to draw a mental map of the Nation around 1900. Measured by the mention of place names, the region west of Houston was deeply rooted in the newspaper and its readership. The East Coast and the Midwest were also present in the imagination of contemporaries. However, the Southwest, the Northwest, and California hardly appear on this mental map. Based on the newspaper's coverage, one could argue that readers of the *Houston Post* around 1900 were barely aware of the Nation as a geographical entity. In economic history, historical newspapers have recently been used to identify treatments or measure variables of interest. Beach and Hanlon (2023) give an overview of the recent use of historical newspaper data in economic history. An interesting recent example is Ferrara et al. (2024), who used digitized newspaper archives to measure a county's exposure to the boll weevil around 1900. The boll weevil is a pest of cotton that hit the American South between 1892 to 1922. The pest reduced cotton production and, consequently, hastened social changes in the primarily Black rural communities, like the fertility transition and higher investment in education.

Even though newspaper portals are an essential source for historians and other disciplines interested in history, such as economics, their potential has not yet been fully realized (Beach and Hanlon, 2023). Firstly, researchers have so far mainly used US-American portals. The reason for this bias may be these portals have been established longer than in other regions of the world. Secondly, the mass utilization of newspaper data is often limited to a keyword search, which usually only covers the entire page and does not discriminate between articles. Therefore, the joint occurrence of two or more search terms is recorded for the page, not the article, and information retrieval is thus still very imprecise (Oberbichler and Pfanzelter, 2021). Thirdly, the text cannot always be downloaded easily, which makes further processing by researchers more difficult. On the other hand, the image files of individual newspaper pages are easy to obtain via the portals. Deep learning algorithms that recognize the layout of a newspaper page and capture the text at the article level, therefore, promise great benefits for historical research. The *Chronicling Germany* data set presented here, comes with layout annotations for every page. It is intended to stimulate the further development of deep learning algorithms and to promote the increased use of non-American newspaper portals.

In addition to more accurate and straightforward information retrieval, downloadable article-level data will also allow scholars of history to apply advanced NLP-methods in the future, including document and text embedding techniques and fine-tuning large language models to 19th-century German.

### A.3 Annotation Guidelines

### A.3.1 Introduction

These annotation guidelines are an adaptation of the OCR-D rules (`https://ocr-d.de/en/gt-guidelines/trans/transkription.html`). We outline additional rules, we created to ensure consistency of the *Chronicling Germany* dataset.

### A.3.2 Page types and type area

The OCR-D guidelines provide for a distinction to be made between page types and the type area during layout analysis. The type area usually contains the text body, but not elements such as the page number. In the *Chronicling Germany* data set, these steps are currently not taken into account.

### A.3.3 Regions

**Region-types** The OCR-D guidelines distinguish between different types of regions, such as text, image and separator regions. In the Bonn Newspaper dataset, the regions are generally recorded in

accordance with OCR-D page region level 1 (`https://ocr-d.de/de/gt-guidelines/trans/ly_level_1_5.html`). However, tables are also recorded as a separate region and no distinction is made between images and drawings; instead, all images, photos, illustrations and drawings are grouped together under the GraphicRegion. The entire contiguous region is always marked as a block. For text regions, this applies to contiguous blocks of the same class, see subsection A.4.

- `TextRegion:` All texts that are not tables. Table headings are not marked as a text region.

- `TableRegion:` All parts of the page that contain tabular information. These are often, but not always, clearly recognizable as tables by small separators. Text that is only separated by separators does not count as a table, but a structure must be recognizable that assigns certain meanings to rows and columns. Table headings are included with corresponding tables.

- `SeparatorRegion:` All dividing lines are marked as SeparatorRegion. This also includes decorative elements that, like other separator lines, separate areas from each other and are not purely cosmetic in nature. The separators are divided into vertical and horizontal separators and marked with "separator_vertical" and "separator_horizontal".

- `GraphicRegion:` All graphics, images, photos, illustrations, and drawings.

### A.4   TextRegion subtypes

TextRegions are divided into different subtypes. The subdivision corresponds to the OCR-D guideline for text regions (`https://ocr-d.de/de/gt-guidelines/trans/lytextregion.html#textregionen__textregion_`). However, drop capitals are treated differently from OCR-D. These are counted as part of the paragraph instead of being marked as a separate text region so that models trained on this data will include them in the correct position in their text output. In addition, headlines (caption) and inverted text (inverted-text) are also recorded in the *Chronicling Germany* data set. Instead of annotating advertisements separately, the classes created for other newspaper pages are applied to the advertisements as far as possible. Because headlines should be visually identified, this leads to a large number of text in the advertisements marked as headlines, which contradicts a semantic definition of a headline. Therefore, it makes sense to treat these pages separately in practice and not differentiate between headings and other text. The following elements from the OCR-D guidelines are not represented in the *Chronicling Germany* dataset due to lack of occurrence: page-number, marginalia, footnote, signature-mark, catch-word, floating, TOC-entry

We discuss the definition for the text subclasses below:

- **paragraph:** Standard text type that includes paragraphs. These are usually kept compact to accommodate as much text as possible in the available space. If a text region cannot be assigned to any other type, it falls under the paragraph label.

- **heading:** Headings that can be clearly distinguished visually from the rest of the text. This is achieved by using a significantly larger or bold font and centered text, which is clearly different from the block layout of paragraphs. A heading is located above a paragraph and is sometimes separated from the previous text by a separator. A thin separator between the heading and the text can occur. However, if there is too much space between them or a thick separator, the two texts no longer count as belonging together in the sense of heading and paragraph. If a text is not superordinate to a paragraph, it cannot be a heading.

- **header:** Page or column titles that appear prominently above the entire page. These are centered at the top of the page and can appear in different font sizes.

- **caption:** Title lines that are located to the right and left of a page heading or text heading. They often contain information such as the date.

- **inverted-text:** Text that is printed white on black. This is often part of decorative elements but is not marked as a graphic element.

## B   OCR

A prerequisite for text recognition is baseline or text-line recognition. Both the baseline itself and a polygon around the text line are annotated. These are generated automatically and only corrected if

the baseline connects non-contiguous text passages. Lines that have been divided into two baselines are not corrected. Tables and inverted text are not given baselines.

The text is corrected according to its optical appearance. What is written on the page is transcribed, even if there are errors in the print or scan. Completely illegible passages are not transcribed.

The transcription is carried out according to level 2 of the OCR-D guidelines (`https://ocr-d.de/en/gt-guidelines/trans/level_2_2.html`). This includes the transcription of special characters such as the 'long s' (U+017F) or long hyphens (U+2014, em dash). Consistency with the rest of the data is important here. As these were generated automatically, it is best to look for another example and adopt that version if the special characters are unclear.

Unlike in the OCR-D guideline, fractions are not transcribed with special characters. Instead, the fraction is represented with a slash:
$1\frac{3}{4}$ = 1 3/4.
In this case, it is important to separate the whole number from the fraction with a space. The same applies to times with an underscore. Example for clock times: $11_{45}$ =11_45 or $11._{45}$ =11._45. (For both, use non-breaking spaces in future (U+202F))

Transkribus allows the selection of special characters with a virtual keyboard. However, it must be ensured that the character used is unique. For example, U+2014 and U+2015 are visually indistinguishable. U+2014 must be used for long hyphens. If the characters are unclear, the OCR-D guidelines, which include tables for the use of special characters, can also be consulted:

- `https://ocr-d.de/en/gt-guidelines/trans/trLigaturen2.html`
- `https://ocr-d.de/en/gt-guidelines/trans/trFremdsprache.html`
- `https://ocr-d.de/en/gt-guidelines/trans/ocr_d_koordinationsgremium_codierung.html`
- `https://ocr-d.de/en/gt-guidelines/trans/trBeispiele.html`
- `https://ocr-d.de/en/gt-guidelines/trans/tr_level_1_3.html`
- `https://ocr-d.de/en/gt-guidelines/trans/trAnfZeichen.html`
- `https://ocr-d.de/en/gt-guidelines/trans/trGedankenstrich.html`

