# OpenReview forum: "Chronicling Germany: An Annotated Historical Newspaper Dataset"
_NeurIPS.cc/2024/Datasets_and_Benchmarks_Track — Submitted to NeurIPS 2024 Track Datasets and Benchmarks_

### Official Review · Reviewer_ZtnH · 2024-07-13
**Chronicling Germany: An Annotated Historical Newspaper Dataset**

**Rating:** 4
**Confidence:** 5

**Review:**

- Introduces a new dataset  and pipeline for historical German-language transcription, contributing significantly to the field.
- No comparison with other German-language datasets, missing context for its contributions.
- Should to follow NeurIPS D&B guidelines, guidelines could be improved and needs reformatting the appendix would enhance readability.

**Strengths:**

- The paper introduces a deep learning-based pipeline and OCR method for extracting texts from historical German-language newspaper images.
- The proposed pipeline addresses multiple tasks, such as layout detection, segmentation, and OCR, offering a nice and  holistic approach to document image analysis.
- The authors introduce a dataset named of historical German-language Newspapers, which is useful for training and evaluating OCR models.
- The evaluation is conducted using both in-domain and out-of-domain test data, providing a good measure of the model's generalization performance. This is the notable contribution of the paper

**Additional Feedback:**

See the above comments

**Clarity:**

The paper lacks clarity and coherence, making it difficult to follow. Some ideas are too
generic.

**Correctness:**

The generic flow of the pipeline presented in the paper looks fairly described, however, most  of the sections about the dataset are not clear and easy to follow.

**Documentation:**

Some information is overlooked, such as:
-  The intended tasks and specific number of instance details of the dataset.
- Texts which are not part of the paper, from the latex template are included e.g in the checklist section

**Ethics:**

Since the dataset is composed from Historical newspaper there might raise an
ethical concerns such as personal information, biases, representative, and cultural
sensitivity.

**Limitations:**

-The authors have not provided the limitations of their work, which leaves room for unforeseen negative impacts. Given that historical documents often contain information about individuals, it is important to acknowledge and address potential ethical and privacy concerns that may arise.

**Opportunities For Improvement:**

- Experimental setup and results section could be rewritten for clarity. For example, it is not clear how many textile images were used for training and testing the OCR model (section 4.3). Sections 4.1 and 4.2 are not clearly stated.
- It would be beneficial for the authors to include the checklist provided in the NeurIPS conference template and provide a proper datasheet. This would make their contribution clearer to the readers.
- Better to have consistency in the number of samples across the document. For example, it is better to present the statistics in the form of a table, detailing the number of pages, text lines, etc.
- The paper covers multiple tasks but fails to mention the number of instances and corresponding ground-truth information for each task.
- The novelty of the presented pipeline and dataset in the paper is questionable to me at least, as there are already numerous open-source Document Image Analysis (DIA) tools and historical German-language OCR datasets e.g  see this paper https://arxiv.org/pdf/1809.05501 and this dataset https://zenodo.org/records/1344132

**Relation To Prior Work:**

The authors haven’t covered related works and datasets in detail and properly.

**Summary And Contributions:**

This paper presents a new dataset of historical German-language newspapers, (approximately 581 scanned newspaper pages from period between 1852 and 1924), along with a pipeline for extracting texts from these newspapers. The pipeline is useful for several tasks, including layout detection and segmentation, as well as OCR. Baseline results for each component of the pipeline are reported, along with the results of the complete end-to-end pipeline.

---

> ### Author Rebuttal · Authors · 2024-08-16
>
> We thank ZtnH for participating in the review process and for the valuable feedback.
>
> 1. _No comparison with other German-language datasets, missing context for its contributions._
>
>     - Thank you for the suggestion. We already discuss the Europeana corpus [r3] in the paper, ist contains 500 pages most of which are in German. With currently well over 300.000 annotated text lines, our Dataset already provides significantly more data than the Europeana dataset with approx. 200.000 text lines.
>     We currently compare to the pipeline from the UB-Mannheim in Table Four of the paper. We will additionally discuss their 197-page "Deutscher Reichsanzeiger und Preußischer Staatsanzeiger" [r1] data set in the related work. Furthermore, we will add the 174 Page Neue Züriche Zeitung [r5] dataset to the discussion of related work. This dataset is larger than both, consequently, we argue that significant progress has been made.
>         - [r3] Europeana corpus - https://ieeexplore.ieee.org/abstract/document/7333898
>         - [r4] Universitätsbibliothek Mannheim - "reichsanzeiger-gt" - https://github.com/UB-Mannheim/reichsanzeiger-gt/
>         - [r5] Universitätsbibliothek Mannheim - Neue Zürcher Zeitung Dataset -  https://github.com/UB-Mannheim/NZZ-black-letter-ground-truth
>
> 2. _Experimental setup and results section could be rewritten for clarity. For example, it is not clear how many textile images were used for training and testing the OCR model (section 4.3). Sections 4.1 and 4.2 are not clearly stated._
>
>     - We kindly refer Ztnh to the "Data" paragraph just under section header four, as well as the "Test-Data" paragraph just under section header five. Both start with the enquoted words set in boldface letters. The data remains the same throughout section 4, therefore the information appears only at the beginning. We will add a short sentence to clarify this point.
>
>
> 3. _It would be beneficial for the authors to include the checklist provided in the NeurIPS conference template and provide a proper datasheet. This would make their contribution clearer to the readers._
>
>    - The checklist is part of the submitted PDF. We kindly ask reviewer ZtnH to check page 16 and onward.
>    - We have added a data sheet for the current state of the dataset in the main rebuttal at the top this page.
>
>
> 4. _Better to have consistency in the number of samples across the document. For example, it is better to present the statistics in the form of a table, detailing the number of pages, text lines, etc._
>
>
>     - We will add more detailed statistics in the paper including information about the origin and time of all pages, number of regions and text lines, as well as the statistics on the dataset split.
>
>
>
> 5. _The paper covers multiple tasks but fails to mention the number of instances and corresponding ground-truth information for each task._
>
>     - We kindly refer Ztnh to the "Data" paragraph just under section header 4, as well as the "Test-Data" paragraph just under section header five. Both start with the enquoted words set in boldface letters. The data remains the same throughout section 4. We will add a short sentence to clarify this point.
>
>
> 6. _The novelty of the presented pipeline and dataset in the paper is questionable to me at least, as there are already numerous open-source Document Image Analysis (DIA) tools and historical German-language OCR datasets e.g see this paper https://arxiv.org/pdf/1809.05501 and this dataset https://zenodo.org/records/1344132_
>
>     - Thank you for pointing out related work. We will cite [r6] in the text. However, we believe this is a miscommunication. Like [r1] our work focuses on the combination of layout and text recognition, supplementary Figure 6 illustrates why this combination is important in practice. To correctly digitize a page we must understand the layout and the text. 1 works with pre-segmented lines of text, which will not solve the structure detection problem. Combined processing of layout and text is a harder task. This dataset allows both in the German language setting.
>     Therefore, our work complements already existing work of international and German historical newspaper datasets. We provide not only more ground truth data for German historical newspapers but also include underrepresented page types like advertisements, lists of fallen soldiers and promotions, as well as stock exchange reports. The layout of these pages differs significantly from standard newspaper pages. Advertisements include challenging line breaks e.g. when the first letter of an advertisement spans over several lines. [r1] focuses exclusively on the Fraktur font in German books. This makes it only viable for improving the plain OCR task for German fraktur font and not for the complete task of transcribing German newspapers.
>
>     - [r6] Ground Truth for training OCR engines on historical documents in German Fraktur and Early Modern Latin, https://arxiv.org/pdf/1809.05501
>     - [r1] American Stories: A Large-Scale Structured Text Dataset of Historical U.S. Newspapers, https://arxiv.org/pdf/2308.12477
>
>
> 7. _The authors haven’t covered related works and datasets in detail and properly._
>
>     - Thank you for suggesting [r6]. This work focuses on both understanding text and layout. Consequently, we discuss mostly other projects like [r1] with this focus. H4zj pointed towards [r7] which does the same for Japanese documents. We will add both [r6] and [r7] to our related work section.
>
>     [r7] Shen et. al 2020, "A Large Dataset of Historical Japanese Documents with Complex Layouts"
>
> 8. _The authors have not provided the limitations of their work_
>
>     - We kindly refer ZtnH to section 6; "Limitations and social impact".
>
>
> 9. _overlooked documentation information - number is instances missing, task unclear checklist missing_
>
>     - The total number of annotated pages (instances) appears in the abstract.
>     - The checklist is part of the submitted PDF from page 16 onwards.

---

> > ### Comment · Reviewer_ZtnH · 2024-08-30
> >
> > Thank you to the authors for addressing most of my comments. I just increase the evaluation score to 5.

---

### Official Review · Reviewer_ffkZ · 2024-07-18
**An Annotated Historical Newspaper Dataset**

**Rating:** 4
**Confidence:** 4
**Clarity:** The paper is well-written

**Review:**

This paper addresses an important gap in available resources by focusing on historical German newspapers. The dataset not only includes annotations for layout analysis but also for text recognition and reading order tasks. The quality of the annotations is commendable, providing detailed and pixel-level labeled data. However, as the first publicly available German historical newspaper dataset, there are a few limitations that need to be addressed. Firstly, the amount of data in the dataset is relatively limited, which may impact its applicability in certain scenarios. Secondly, the authors did not perform extensive benchmark experiments or evaluate the performance of existing models on the dataset. Additionally, they did not analyze the specific challenges posed by the dataset. Therefore, further revisions are necessary to enhance the paper. The authors should conduct more experiments and analyses to validate the effectiveness of the proposed dataset and provide a more comprehensive evaluation.

Quality: The quality of this dataset is exceptionally high, primarily due to the extensive manual annotations performed by historical experts. The dataset includes detailed layout information such as the precise locations of text blocks, advertisements, and images, as well as annotations for text lines and reading order, which are crucial for accurate OCR and text processing. Detailed annotations have been performed on 581 pages, including 1,900 individual advertisements and 26,000 layout polygon regions, ensuring the reliability and accuracy of the data. Additionally, historical newspapers serve as valuable resources for studying social groups' opinions and cultural values, further enhancing the dataset's value. However, a notable drawback of this dataset is the relatively limited amount of data.

Clarity: The paper is well-organized and clearly written, providing a detailed explanation of the dataset's creation and annotation process. The methodology section is thorough, clearly explaining the processing pipeline and its components. The authors effectively convey the importance and potential applications of the dataset, making the paper highly readable to a broad audience, including those outside the fields of digital history or computer vision. However, as a benchmark dataset, the paper lacks sufficient experiments and performance analysis using existing models, which affects its reliability and comprehensiveness. Additionally, there is a lack of in-depth analysis of the dataset's challenges.

Originality: This dataset is unique in its focus on historical German newspapers, a previously under-explored area in the field of digital history. Most existing datasets and OCR systems are designed for modern documents or English-language historical texts, which differ significantly in layout and typography. The Chronicling Germany dataset addresses this gap by providing a specialized resource that captures the unique challenges posed by historical German newspapers, such as the use of Fraktur font and the archaic 'long s'. This originality not only fills a significant gap in the available resources but also opens up new avenues for research in historical document processing and low-resource tasks in computer vision.

Significance: The dataset and the accompanying processing pipeline have substantial implications for both digital history and low-resource computer vision tasks. For digital historians, this dataset provides a valuable tool for studying social groups' opinions and cultural values from the past, offering insights that were previously difficult to obtain due to the lack of reliable layout information. For the computer vision community, the dataset serves as a benchmark for developing and testing OCR systems and other document processing tools tailored to historical documents. The successful establishment of a baseline recognition pipeline further demonstrates the dataset's practical utility, paving the way for future research and development in both fields. The public availability of the dataset and code promotes transparency, reproducibility, and collaboration among researchers, amplifying the dataset's impact.

**Strengths:**

1. This dataset fills a significant gap in resources for historical German newspapers, allowing researchers to study the perspectives and cultural values of social groups over time. These newspapers differ significantly from modern documents and other historical datasets.

2. The dataset is of high annotation quality, including detailed layout information such as the precise locations of text blocks, advertisements, and images, as well as annotations for text baselines, text recognition, and reading order. This is crucial for accurate OCR and text processing.

3. The dataset and code are publicly available, promoting reproducibility and further research.

**Additional Feedback:**

None

**Correctness:**

The claims and arguments presented in the paper are correct, and the dataset construction method is sound. However, as a benchmark dataset, the evaluation methods and experimental design in this paper are inadequate. The paper lacks several essential experiments, it does not use existing document layout analysis and text line detection models on this dataset, lacks performance and visualization analysis, and fails to provide subsequent researchers with an overview of the current state of development. Additionally, there is no analysis of the problems that existing models cannot solve on this dataset, nor a detailed analysis of the dataset's challenges.
For the layout analysis task, the paper only uses IoU as the evaluation metric. However, commonly used metrics also include recall, precision, AP50, AP75, and mAP.

Therefore, the current version of this paper still requires significant revisions. The authors should conduct more experiments and analyses to provide a more detailed and reliable benchmark, and to validate the effectiveness of the proposed dataset.

**Documentation:**

The dataset documentation is detailed, covering data collection, organization, availability, which supports reproducibility.

**Ethics:**

There are no ethical concerns.

**Limitations:**

The authors have thoroughly discussed the limitations of their work, and there are no potential negative societal impacts associated with this paper.

**Opportunities For Improvement:**

1. The dataset has a relatively limited amount of data.

2. The related work section lacks an introduction to existing layout analysis and document detection and recognition models.

3. As the first German layout analysis dataset, I suggest the authors should conduct a more comprehensive benchmark experiment. This should include an analysis of the performance of state-of-the-art methods from 2022-2024 on this dataset, along with model comparisons and visual analyses. This would provide subsequent researchers with a perspective on the current state of development. Additionally, the authors should analyze the unresolved challenges within the dataset to facilitate further research by others.

4. The description of the annotation process in the paper is relatively brief. More detailed documentation and guidelines on the annotation process could help other researchers replicate this study.

**Relation To Prior Work:**

The paper clearly discusses how this work differs from and improves upon previous contributions, particularly in the context of historical German newspapers.

**Summary And Contributions:**

This paper introduces the Chronicling Germany dataset, which addresses the issue of missing layout information in historical German newspapers. The dataset comprises 581 annotated pages from 1852 to 1924, annotated by historical experts. The paper outlines a processing pipeline and establishes baseline results for both in-domain and out-of-domain test data. The dataset and baseline code are publicly available, providing a foundation for future research in digital history and low-resource computer vision tasks.

---

> ### Author Rebuttal · Authors · 2024-08-16
>
> We thank ffkZ for the valuable Feedback.
>
> 1. _The dataset has a relatively limited amount of data._
>
>     - We could not agree more, since a bigger dataset is always better. Please bear in mind that we are talking about pages densely packed with text. At the time of submission, 1,500 hours of annotation time had already been invested. As our annotation efforts continue, we have been able to make significant extensions to the original submission. We also made this point in the reply to H4zj. In detail, we added 88 pages to the generalization part of the dataset. In addition, we plan to add 50 extra pages on top of these before the conference deadline.
>
>     - This dataset was already large at submission time compared to prior work with a comparable focus on European sources. We already compare our work to the Europeana dataset [r3] (500 Pages), in the submission. We will additionally mention the Deutscher Reichsanzeiger [r4] (197 Pages) and the Neue Züriche Zeitung [r5] (174 Pages) sets in the related work. With currently well over 300.000 annotated text lines, our Dataset already provides significantly more data than the Europeana dataset with aprox. 200.000 text lines. The Europeana Dataset includes predominantly German pages, as well as a couple of other European languages. This 581 page work consequently constitutes a significant extension of existing work.
>
>         - [r3]  Europeana https://ieeexplore.ieee.org/abstract/document/7333898
>         - [r4] Deutscher Reichsanzeiger https://github.com/UB-Mannheim/reichsanzeiger-gt/tree/main
>         - [r5] Neue Züriche Zeitung https://github.com/UB-Mannheim/NZZ-black-letter-ground-truth
>
> 2.  _The related work section lacks an introduction to existing layout analysis and document detection and recognition models._
>
> - We kindly refer ffkz to section 2.1 which presents an introduction to document processing pipelines such as the one from Dell et al. as wel as section 2.2 where pipeline elements are presented in more detail. We have included the most important models we are aware of. If you suggest additional models, we will cite those as well.
>
> 3. As the first German layout analysis dataset, I suggest the authors should conduct a more comprehensive benchmark experiment. This should include an analysis of the performance of state-of-the-art methods from 2022-2024 on this dataset, along with model comparisons and visual analyses. This would provide subsequent researchers with a perspective on the current state of development. Additionally, the authors should analyze the unresolved challenges within the dataset to facilitate further research by others.
>
>   - Thank you for the pointer. We have evaluated the pipeline provided by Dell et al. [r1]. We evaluate the pipeline first without fine-tuning. This produced satisfactory results on the layout, achieving an average IoU of 0.817 IoU, even though the annotations of the two datasets are not always consistent. We have included an example image in the rebuttal PDF. On the other hand, the antiqua-pretrained OCR model did not generalize well to the "faktur" texts. We observe an average Levensthein distance of 0.589 on the test set and 0.574 on the generalization dataset.
> We are currently working on fine-tuning the OCR to provide a fairer comparison between the two pipelines. We will report back when we know more.
>
>   - We also evaluated the layout parser pipeline [r2]. The layout parser comes with a newspaper-navigator pipeline, which is pre-trained on historical newspapers. Unfortunately, a text label is unavailable, so we cannot use this model to detect text.
>   (See also https://layout-parser.readthedocs.io/en/latest/notes/modelzoo.html#example-usage . The label map for the "NewspaperNavigator" does not include a text label.) The remaining networks are not trained on historical newspapers.
>
>     - [r1] American stories: A large-scale structured text dataset of historical us newspapers - https://proceedings.neurips.cc/paper_files/paper/2023/file/ffeb860479ccae44d84c0de32acd693d-Paper-Datasets_and_Benchmarks.pdf - https://github.com/dell-research-harvard/AmericanStories
>     - [r2] LayoutParser: A Unified Toolkit for Deep Learning Based Document Image Analysis - https://arxiv.org/abs/2103.15348
>
>
>   - We will specifically add unresolved challenges in section 6 by highlighting layout or OCR elements that are present and correctly annotated in the dataset but remain challenging to transcribe. In addition to the Fraktur font, this applies most notably to the drop-capitals present in advertisement pages, currently described in section 1. Furthermore, financial and stock exchange reports contain abbreviations and fractions that are hard to transcribe.
>
>
> 4. _The description of the annotation process in the paper is relatively brief. More detailed documentation and guidelines on the annotation process could help other researchers replicate this study._
>
>     - Thank you for the feedback. Supplementary section A.3 presents our annotation guidelines in detail. We will make this more apparent in the main body of the text.
>     - Footnote 5 on page 3 provides links to both the baseline pipeline code and the dataset, which allows the complete reproduction of this work.
>
>
> 5. _commonly used metrics also include recall, precision, AP50, AP75, and mAP._
>
>   - We are working on reporting numbers with these metrics as well. We will report back to you, when we know more.

---

> > ### Comment · Reviewer_ffkZ · 2024-08-31
> >
> > The authors have provided explanations addressing the five main concerns I raised. I find their responses of point 2, 4 and 5 satisfactory, with the exception of point 3 where I feel the reply could have been more comprehensive. Taking into account their explanations and the opinions of other reviewers, I would like to raise my evaluation score to 5, but I cannot find any button/menu to revise my score.

---

### Official Review · Reviewer_H4zj · 2024-07-22
**Pipeline and dataset of historical German/Cologne newspaper page layouts**

**Rating:** 5
**Confidence:** 5
**Clarity:** The paper is clearly written and very…

**Review:**

"Chronicling Germany" is well written and clear presents its vision. The paper outlines the pipeline to curate a dataset of German (predominantly Cologne) newspaper page layouts.

Model components of the pipeline, accuracy metrics (e.g. CER) to assess OCR capabilities, and its limitations are - for the most part - reasonably motivated. The work is original in the dataset it tackles: Historical German newspapers use the Fraktur font and are of particularly dense layout. The authors correctly identify the need to augment pre-existing work that largely focuses on US newspapers in this scientific sub-domain. The work is also significant as a sizable amount of expert labor is deployed to annotate hundreds of these dense newspaper pages.

However, there is a chance for improvement w.r.t. the temporal and geographical scope of the sourced newspapers (mostly Cologne; 1866), the diversity of the test set, and minor clarifications in the paper (e.g. Nougat, HJD and WordScape references). When sufficiently addresses a considerably wider reach for this work is to be expected.

**Strengths:**

1. [Intricacies of the dataset]. The authors establish the challenging nature of this historical newspaper dataset. They effectively demonstrate the potential value of such a dataset by highlighting key attributes that deviate from similar pre-existing newspaper datasets. Notable distinctions include the predominant use of the Fraktur font, the presence of special characters (e.g., the archaic "long s" and the contemporary 'ß'). Overall, the dataset contains approximately 26K polygons. These annotations are (presumably) of high quality due to the use of domain experts. The authors committed to publish the data (annotations).

2. [Visual presentation of data characteristics] The authors use tables and figures effectively to provide an overview of what the data entails.

3. [Well-crafted processing pipeline] The processing pipeline is presented in a highly descriptive manner. It appears the authors made an extensive effort in compiling the model components - referring to and being informed by state-of-the-art models in the domain of document layout understanding in digital history. They provide crucial technical details (e.g. input image resolution and hyperparameters for data augmentation/model training), ensuring the work's reproducibility. Additionally, the authors have committed to making the code publicly available.

4. [Historical context]. The historical context is well-presented for readers that are not overly familiar with German newspapers of the 1860s to 1920s.

5. [Presentation and Style] The paper is very well written.

**Additional Feedback:**

This has the chance to be a great paper if the suggestions for improvements are sufficiently addressed.
I will adjust my rating based on the degree to which these points are addressed. I hope you perceive this as encouragement.

**Correctness:**

The abstract is very concise and well-written. However, the statement 'layout information is often missing' may not be appropriate as the related work cited seems to provide bounding boxes for document layout elements (e.g., 1.14B for Dell et. al (2024)). While “1,500 hours annotating” indicates a valiant effort it might be more informative to the reader to highlight the actual origin of the data (mostly ‘Kölnische Zeitung’ and 1866/1924).

Section 2 “Related Work” does a formidable job in outlining OCR/layout detection efforts. “Surprisingly, their network generalizes to old mathematical literature. However, Arxiv papers do not resemble historical newspapers”. While Nougat is predominantly trained on ArXiV PDFs and Vision Transformers tend to be bridle for out-of-distribution-generalization the authors extensively highlight data augmentation techniques that facilitate demonstration. “Notably” might be more appropriate.

*“The text correction process is ongoing. To date, we have 124 corrected pages. All pages will be ready in time for the Conference in December.”* Please elaborate on how correctness is ensured. E.g. will multiple annotators cross-check an annotated page?

**Documentation:**

Data on the annotation _guidelines_ are provided. Data (once correction is completed) and code will be provided in the future according to the authors. Reproducibility is therefore not entirely ensured at the current point in time.

**Ethics:**

No ethical concerns. The newspaper pages are already publicly available online.

**Limitations:**

While limitations are present in the paper and outlined in "Opportunities For Improvement" the authors make a good attempt in highlighting them. The lack of historical events captured by the dataset (at the current point of submission) should be elaborated on, however.

**Opportunities For Improvement:**

This work is very intriguing. Regardless, it offers several avenues for improvement to increase factuality and bolster its reach.

1. **Limited time/scope of the (annotated) dataset** Naturally, every scientist wishes their dataset to be of larger sample size or at least sampled more diversely. This is particularly true for low-resource CV tasks where historical data is scarce and the labor of human experts to annotate it costly. Regardless, a dataset that consists of 95% of page images sourced from a single newspaper (Kölnische Zeitung/Cologne Newspaper) from two years (1866 and 1924) hardly chronicles Germany. While the ‘Kölnische Zeitung’ appears to be one of the major German newspapers of the time as well as 1866 and 1924 being important years - many stretches of 1852-1888 such as the Franco-Prussian War (1870/1871), the proclamation of the German Empire (1871), the era post-Press Act (1874) touted by the authors are underrepresented with a mere 24 annotated pages. Events between 1889 and 1923 including the Wilhelmine Period (1900-1918), World War I (1914-1918/19), the creation of the Weimar Republic (1919) and its early crises (Hyperinflation in the Weimar Republic and the Beer Hall Putsch (1923)) are excluded altogether. This might be an issue beyond historical significance as Dell et al. (2024) report drastic variations in CER across decades (8.9% in 1850s to 1.8% in 1910 potentially highlighting consistently higher error rates for older newspaper pages). The current data does not allow a fine-grained evaluation casting doubts on the generalization ability of models trained on it. The following items would address these issues: extension of the annotated dataset in sample size and scope (newspapers and publication years); in particular a (a) significant increase of the out-of-distribution test set that is (b) sampled uniformly in time from 1852-1924 across (c) diverse newspapers as the processing pipeline evaluation hinges on this quality

If not sufficiently addressed it might be worthwhile considering to rebrand the work as "Chronicling the Rhineland (during the Second German Empire)". While less ambitious than "Chronicling Germany" it is more accurate and likely attracts the actual audience it is targeting. There were a handful of "Germanies" and an abundance of historical events not captured in the data (at this point).

2. **Processing Pipeline**: The second pillar of this work is to establish baseline results for the processing pipeline. While key metrics such as IoU and CER are reported, only a single reference pipeline is evaluated ‘UB Mannheim (2024)’. This limits the strength of the empirical evaluation. If the “Chronicling Germany” dataset grants its own processing pipeline it begs the question how a different pipeline (designed for Antiqua/English but trained on “Chronicling Germany” data) would stack up.

3. **Model evaluation on the test set** The following statement in 4.4 reads: “For each component, we choose the model with the best results and use the result of each component for the next one” gives the impression that the components are chosen on the test set itself (rather than the train set). If this is indeed the case it will lead to deflated character error rates. Please ensure that the test data is truly held out and describe the procedure in subsection 4.4 more clearly to prevent confusion.

4. **Choice of metrics to assess OCR** Some of the cited works (e.g. Nougat by Blecher et al. (2023)) evaluate accuracy of the OCR task on a word-by-word basis through metrics like BLEU or ROUGE. Elaborating why the CER (character error rate) is chosen is likely to extend the reach of this work to a wider audience.

5. **Transfer learning or lack thereof in Table 2** The authors compare layout detection capability via IoU for ImageNet- and Europeana-pre-trained weights that show little difference in performance - except for caption and separators. Is this due to the unusual size of the separator bounding boxes? Highly relevant: WordScape (Weber et al., 2023) conducted experiments that measure the usefulness of the pre-training as a function of downstream dataset size that might elucidate this experiment further. Mildly relevant: Other works in image segmentation have a more lively debate on what exactly the advantage of domain dataset-specific pre-training vs. ImageNet-pre-trained weights is (Wen et al., 2021 "Rethinking pre-training on medical imaging").

6. **Historical Japanese Dataset** In terms of culturally diverse, historical layout annotation datasets HJD is quite impactful: Shen et. al 2020, "A Large Dataset of Historical Japanese Documents with Complex Layouts". I think citing it is advisable.

**Relation To Prior Work:**

Mostly yes. In particular the other works on historical layout detection are well presented.
Adding HJD (Historical Japanese Dataset) and Wordscape to the citations will improve it further.

**Summary And Contributions:**

This paper presents a dataset of historical German newspapers with highly dense and therefore complex layouts; including 581 high-resolution, human-annotated newspaper pages predominantly sourced from the Kölnische Zeitung in the year 1866. Additionally, the authors present a model pipeline that extracts layout information (9 classes excl. background) and conducts OCR on such historical newspaper page images. The authors validate this processing pipeline on two test sets against a baseline reporting object detection and OCR accuracy via IoU (intersection over union) and CER (character error rate) among other metrics.

---

> ### Author Rebuttal · Authors · 2024-08-16
>
> We thank reviewer H4zj for the excellent feedback. We are quite impressed with the quality of the review. We are addressing every point in detail. Citations from the review are set in italics.
>
> 1. _Limited time/scope of the (annotated) dataset Naturally, every scientist wishes their dataset to be of larger sample size or at least sampled more diversely._
>
>   - We have seized the past few months to expand our dataset, focusing on generalization. It now contains 88 additional pages of both manually annotated layout and text (corrected OCR) with $\approx$ 24.000 additional lines of text and $\approx$ 3300 additional layout polygons. Please check the main rebuttal for more details on the current state of the full dataset.
>
> -  Following your suggestions, we aim to expand our dataset further, annotating layout and text on (at least) another 50 pages:
>
>       | Year | Paper  |  Pages | Reason for inclusion                                                           |
>       |------|---------|--------|----|
>       | 1700 | Reichs-Post-Reuter | 8  | This is the oldest Newspaper from Germany currently available in digital form.|
>       | 1813 | Königlich privilegierte Stuttgarter Zeitung  | 4  | Battle of Leipzig.    |
>       | 1849 | Hamburgischer unpartheiischer Correspondent  | 4 | Frankfurt Constitution. |
>       | 1870 | Weisseritz-Zeitung  | 8  | French war declaration and war preparations. |
>       | 1871 | Karlsruher Zeitung  | 4  | the Imperial proclamation in Versailles.    |
>       | 1898 | Dresdner Journal  | 8  | the Flottengesetz.    |
>       | 1917 | Muenchner Neue Nachrichten | 6  | (US-) Americas war declaration  |
>       | 1918 | Darmstädter Zeitung  | 4  | resignation of the emperor and chancellor, and the new government formed by Friedrich Ebert (later referred to as Weimar Republic). |
>       | 1923 | Eibenstocker Tagblatt      | 4  | Beer Hall Putsch and the planned Rentenmark and Hyperinflation. |
>
>       The time from 1924 onwards constitutes a natural end since newspapers gradually started using Latina fonts instead of fraktur during that period. We aim to complete these annotations by November.
>
> 2. _rebrand the work as "Chronicling the Rhineland (during the Second German Empire)"._
>
>   - This would be a very good idea with the original dataset. However, with our recent extensions, we are uncertain if changing the title is still necessary. We now include more historical events and newspapers. We plan to expand the dataset in this direction.  We would, however, also be willing to consider changing the title to "Chronicling the Rhineland" (and replace some of the planned annotations with Rhenish newspapers).
>
> 3. _pipeline evaluation_
>
>    - Thank you for the pointer. We have evaluated the pipeline provided by Dell et al. [r1].
>      We evaluate the pipeline first without fine-tuning. This produced satisfactory results on the layout,
>      achieving an average IoU of  0.797 IoU, even though the annotations of the two datasets are not always consistent.
>      We have included an example image in the rebuttal PDF. When evaluating the layout detection, we observe the following per class IoU values:
>
>     - | background | caption |  header | paragraph | heading | table |
>       |---------|----|-----|-----------|---------|-------|
>       | 0.703 |  0.0  |  0.071  |  0.838 |  0.461  | 0.390 |
>
>     - On the other hand, the antiqua-pretrained OCR model did not generalize well to the "faktur" texts.
>     We observe an average Levensthein distance of 0.589 on the test set and 0.574 on the generalization dataset.
>    - We are currently working on fine-tuning the OCR to provide a fairer comparison between the two pipelines.
>
> - We also evaluated the layout parser pipeline [r2]. The layout parses comes with a newspaper-navigator pipeline, which is pretrained on historical newspapers. Unfortunately, it does not allow text to be detected.
>
>   - [r1] American stories: A large-scale structured text dataset of historical us newspapers - https://proceedings.neurips.cc/paper_files/paper/2023/file/ffeb860479ccae44d84c0de32acd693d-Paper-Datasets_and_Benchmarks.pdf
>   - [r2] LayoutParser - https://arxiv.org/abs/2103.15348
>
> 4. _Model evaluation on the test set._
>
>   - Thank you very much for the pointer! All components have been selected using 30 pages of the separate validation set. The generalization/test data from section five has had no impact on the model selection. We will rework the section accordingly.
>
> 5. _Choice of metrics_
>
>   - Bleu results on the test and generalization set from time of submission:
>
>     | Pretrained UB Mannheim | From scratch    | Generalization |
>     |---------|-----------------|----------------|
>     | 0.827 +- 0.0024  | 0.823 +- 0.0021 | 0.508 |
>
>     - We suspect word-by-word comparison reduces the generalization score because missing spaces, for example, can break word-by-word comparison, while the Levenshtein distance is more robust.
>
> 6. _Transfer learning_
>
>   - Thank you for suggesting both references. We will discuss both. We believe the differences stem from differing annotation granularity and dataset size.
>
> 7.  _Historical Japanese Dataset_
>
>   -  We will add the citation.
>
> 8. _Availability of data and code_
>
>   -  See footnote 5 on page 3.
>
> 9. _“will multiple annotators cross-check an annotated page?_
>
>   - For the moment, an automatic transcription is checked and corrected by a single human domain expert. Currently, we have corrected 446 pages in of the Kölnische Zeitung, and 112 pages in the generalization part of the dataset. To improve quality further we will run a second correction round, where all lines will again by proofread by different annotators.
>   - We have already performed cross-checking for Layout annotations on the ‘Kölnische Zeitung’.
>
> 10. _'layout information is often missing'_
>
>   - We apologize. We meant to express that layout information is often missing for German language historical newspapers. We will clarify.

---

> > ### Comment · Reviewer_H4zj · 2024-08-27
> >
> > > We have seized the past few months to expand our dataset.
> >
> > This improvement is appreciated. I believe the geographical and temporal distribution is way more diverse.
> >
> > > This would be a very good idea with the original dataset. However, with our recent extensions, we are uncertain if changing the title is still necessary
> >
> > Since the current version of the dataset is more diverse I'd keep the the title as is.
> >
> > > 3. pipeline evaluation
> >
> > I consider these improvements.
> >
> > Thank you for the comments regarding the remaining points. I will take them into account accordingly.

---

### Author Rebuttal · Authors · 2024-08-16

Dear reviewers,

Thank you for the constructive comments and insightful feedback. We were happy to read that reviewers found this dataset challenging (H4zj), of high quality (ffkZ, H4zj), publicly available (ffkZ) and making a significant contribution (Ztnh, ffkZ). Overall, reviewers found our paper well-written (H4zj, ffkz) and organized (ffkZ). Regarding the baseline transcription pipeline, H4zj found the design choices reasonably motivated.

Naturally, our reviewers voiced concerns and saw several avenues for improvement. We summarize the most important points here.

- *Datasheet* We thank *ZtnH* for the suggestion and provide a data sheet for the dataset in its current state (2024.08.16):

  | Time Period   | Newspaper                            | pages  | Lines     | Words     | Region Polygons |
  |---------------|--------------------------------------|--------|-----------|-----------|-----------------|
  | 1785  | Schwäbischer Merkur   | 11     | 1,046     | 7,093     | 91    |
  | 1813  | Donau Zeitung    | 4      | 334       | 1,785     | 32              |
  | 1834  | Fränkischer Kurier   | 4      | 412       | 3,128     | 78              |
  | 1851  | Ostpreussische Zeitung  | 4      | 1,498     | 9,964     | 177   |
  | 1856  | Der Bazar   | 11     | 1,339     | 10,262    | 118             |
  | 1857  | Berliner Börsen Zeitung   | 5      | 1,736     | 8,370     | 196   |
  | 1866  | Bonner Zeitung   | 4      | 1,652     | 10,667    | 264  |
  | 1866  | Neue Berliner Musikzeitung  | 8      | 1,058     | 7,850     | 72   |
  | 1866  | Fränkischer Kurier     | 6      | 1,826     | 12,241    | 224  |
  | 1866  | Pfälzer Zeitung   | 4      | 1,156     | 7,783     | 179     |
  | 1866  | Vossische Zeitung    | 26     | 5,782     | 35,221    | 921  |
  | 1866  | Weisseritz-Zeitung   | 8      | 833       | 5,680     | 143   |
  | 1866  | Kölnische Zeitung | 416    | 249,618   | 2,106,539 | 17,054          |
  | 1867  | Hannoverscher Courier     | 4      | 2,012     | 13,590    | 317             |
  | 1867  | Neue preussische Zeitung   | 4      | 3,066     | 14,625    | 229             |
  | 1852-1888  | Special editions Kölnische Zeitung   | 24     | 969       | 9,605     | 165             |
  | 1891 | Bonner Zeitung | 4  | 1,648     | 9,757     | 422             |
  | 1924   | Kölnische Zeitung  | 141    | 79,832    | 668,574   | 9,067           |
  | --------- | ------------------- | ------ | --------- | --------- | --------------- |
  | Sum           |     | 688    | 355,817   | 2,942,734 | 29,749          |

- *Dataset size and its relation to related datasets* (Ztnh, ffkZ)
  - We have added 88 additional pages since the original submission.
  - We will be able to add 50 additional pages until November, taking H4zj's suggestions into account. See the response to H4zj for further details and the rationale behind the selection.
  - The paper already discusses Europeana ( https://ieeexplore.ieee.org/abstract/document/7333898 ) with its 500 pages from European sources. We will additionally discuss the 197-page "Deutscher Reichsanzeiger und Preußischer Staatsanzeiger" ( https://github.com/UB-Mannheim/reichsanzeiger-gt/ ) data set in the related work. Furthermore, we will add the 174-page Neue Züriche Zeitung ( https://github.com/UB-Mannheim/NZZ-black-letter-ground-truth ) dataset to the discussion of related work. This dataset is larger than both. Consequently, we argue that significant progress has been made.

- *Geographic distribution* (H4jz)
    -  The rebuttal PDF includes a figure outlining the geographic location of the additional papers papers in historic Germany.

- *Access to data and code*: Not all reviewers saw footnote five on page 3. The baseline code and dataset are already available online. We continuously update the dataset as more annotated pages become available.

- *Comparsion to additional pipelines* (ffkZ, H4jz)
    - We are adding an additional comparison to the processing pipeline from [r1].
    We have already finished a comparison to the original model trained on American antiqua-pages.

    - The model produced satisfactory results on the layout,
     achieving an average IoU of  0.797 IoU, even though the annotations of the two datasets are not always consistent.
     We have included an example image in the rebuttal PDF.
     When evaluating the layout detection, we observe the following per class IoU values:

    - | background | caption |  header | paragraph | heading | table |
      |------------|---------|---------|-----------|---------|-------|
      |    0.703   |  0.0    |  0.071  |  0.838    |  0.461  | 0.390 |

    - On the other hand, the antiqua-pretrained OCR model did not generalize well to the "faktur" texts.
    We observe an average Levensthein distance of 0.589 on the test set and 0.574 on the generalization dataset.

   - We are currently working on fine-tuning the OCR to provide a fairer comparison between the two pipelines. We will report back when we know more.


- *Additional metrics* (H4jz, ffkZ)

    - We have already computed BLEU scores on the test and generalization set from the time of submission:

        | Pretrained UB Mannheim | From scratch    | Generalization |
        |------------------------|-----------------|----------------|
        | 0.827 +- 0.0024        | 0.823 +- 0.0021 | 0.508          |

    - We speculate that the reason for the low generalization score lies in the word-by-word comparison. With word-by-word comparisons, falsely identified spaces can have a large impact because these cause words to be mismatched. With the Levenshtein distance, this is not the case.

We thank everyone for their feedback. We believe your suggestions have helped improve this paper. If you feel the same way, please consider raising your score.

[r1] American Stories: A Large-Scale Structured Text Dataset of Historical U.S. Newspapers, https://arxiv.org/pdf/2308.12477

---

> ### Author Response · Authors · 2024-08-30
> **Additional numbers as requested by H4zj and ffkz.**
>
> Dear reviewers,
>
> We understand the desire to see an extended evaluation of the presented pipeline. For the moment, while we can evaluate the pipeline by Dell et al. [r1] on our dataset, we did not annotate the ground-truth text boxes that would be required to fine-tune this pipeline as requested by H4zj.
> The historic newspaper community works with baseline detection or direct text object detection pipelines. [r1] works with a Yolo v8 to detect text objects. Following Kodym et al. ( https://arxiv.org/pdf/2102.11838 , see Fig. 2 ), we employ a U-Net to detect text baselines in this work. Our annotations are consistent with the Europeana-corpus from Clausner et al. (http://schema.primaresearch.org/www/assets/papers/ICDAR2015_Clausner_ENPDataset.pdf) and the work from UB-Mannheim that also features fraktur letters. This choice allows combining our datasets in future work. This is a key design decision since we aim to boost performance in the fraktur-subset of historical newspapers.
> Unfortunately, our choice complicates fine-tuning the pipeline proposed in [r1]. To add an additional comparison, within the rebuttal period, we fine-tuned the OCR transformer proposed at https://github.com/DCGM/pero-ocr/tree/master/pero_ocr/ocr_engine. It appears as transformer (new) in the table below. The transformer performs slightly better than the LSTM-based OCR that was part of the original submission.
>
> |  Model              | Levenshtein-Distance | completely correct [%] | many errors [%] |
> |---------------------|----------------------|------------------------|-----------------|
> |  UB Mannheim (2024) | 0.020                | 47.1                   | 6.3             |
> |  LSTM               | 0.016 ± 0.0013       | 69.140 ± 0.352         | 5.146 ± 0.318   |
> |  transformer (new)  | 0.009                | 69.7                   | 3.77            |
>
>
> In response to reviewer ffkz, we added object level recall and f1 score test set evaluations, for the network trained by fine-tuning the Europeana layout detection weights. Objects are extracted by computing polygons from the pixel-level segmentation maps. We consider a detection as correct if we observe an IoU > 0.5.
>
>
> | Class                 |    count in ground truth  | Precision (mean(median) ± std)  | Recall (mean(median) ± std)   | F1 Score (mean(median) ± std) |
> |-----------------------|---------------------------|---------------------------------|-------------------------------|-------------------------------|
> | caption               | 80                        | 0.36455     (0.32545) +- 0.0774 | 0.666175  (0.5945)  +- 0.1509 | 0.432075 (0.3809) +- 0.1033   |
> | table                 | 154                       | 0.84025     (0.848) +- 0.0256   | 0.808325  (0.80965) +- 0.0217 | 0.791175 (0.7889) +- 0.0134   |
> | paragraph             | 1009                      | 0.907975    (0.9104) +- 0.0185  | 0.94055   (0.93655) +- 0.0137 | 0.918725 (0.91915)+- 0.0031   |
> | heading               | 598                       | 0.83385     (0.83315) +- 0.0192 | 0.88445   (0.8893)  +- 0.0097 | 0.839775 (0.8386) +- 0.0092   |
> | header                | 37                        | 0.969425    (0.9738) +- 0.0153  | 0.9717    (0.972)   +- 0.0140 | 0.963325 (0.96855)+- 0.0177   |
> | separator_vertical    | 308                       | 0.868725    (0.88225) +- 0.0321 | 0.84545   (0.8484)  +- 0.0166 | 0.850225 (0.85855)+- 0.0219   |
> | separator_horizontal  | 811                       | 0.876525    (0.8782) +- 0.0121  | 0.904475  (0.9053)  +- 0.0046 | 0.8879   (0.88915)+- 0.0089   |
> | image                 | 14                        | 0.4738      (0.4762) +- 0.0443  | 0.71855   (0.71855) +- 0.0371 | 0.416075 (0.42025)+- 0.0443   |
> | inverted_text         | 3                         | 0.39165     (0.36665) +- 0.0682 | 0.666675  (0.6667)  +- 0.1178 | 0.3988   (0.38095)+- 0.0702   |
>
> In terms of performance, we see roughly the same picture we saw for the initial evaluation using intersection over union.

---

### Comment · Area_Chair_45xV · 2024-08-28
**Important steps before the end of the discussion period**

Dear Reviewers, thank you for your constructive comments. If you have not done so already, please review and respond to author rebuttals to your review. At a minimum, please acknowledge that you have received the rebuttal, indicate whether/why the points you raised in your original review have/have not been addressed, and provide some reasoning.

If you have already acknowledged author rebuttals, any further engagement is at your discretion.

The discussion period is set to end this Saturday, Aug. 31, and for our purposes the deadline will be Anywhere on Earth.

---

> ### Author Response · Authors · 2024-08-31
> **Thank you for considering raising your score**
>
> Many thanks to the reviewers for their responses to our rebuttals. We are happy that we could address your concerns. Both ZtnH and ffkz have announced that they will increase their score. ffkz writes that ffkz cannot find the button to do so. If you press the edit button on your original review, you should be able to change the score officially.

---

> > ### Comment · Reviewer_ffkZ · 2024-08-31
> >
> > I have double checked but still cannot find any edit button anywhere which can revise or input my final score.  In other conferences hosted on OpenReview, there's usually an Edit button next to my previous reviews. However, up until now, I haven't seen a similar Edit button in the review interface for the NeurIPS 2024 Dataset Track.  I've also noticed that that other reviewers who have stated they want to modify their scores also seem to have not yet changed their scores.

---

### Public Comment · ~Moritz_Wolter1 · 2025-06-13
**Thank you for your Feedback**

We want to thank everyone for their feedback. The final version of this paper is now available at https://openreview.net/pdf?id=YHDfqvtye9  .

---

### Decision · Program_Chairs · 2024-09-26

**Decision:**

Reject

**Comment:**

This paper presents a novel dataset of images of meticulously annotated historical, German-language newspapers. The dataset represents a significant investment of effort. Both layout and text have been annotated. The dataset is challenging for both humans and machines to annotate because it includes archaic characters, is set in a typeface not found in modern datasets, and includes difficult layout elements like stock market reports and advertisements.

Reviewers (H4jz, ffkZ) appreciate the contribution to a low-resource problem and the historical significance of the archives. Your AC should add that many social scientists of a comparative or historical bent in disciplines beyond history will appreciate the contribution of both machine-readable text and a reference implementation of an OCR pipeline tailored to this period of German history. it is of particular interest to studies of state formation and politics.

Reviewers (H4zj, ZtnH) also praise the description and engineering of the data processing pipeline as an exemplar.

All reviewers question dataset size and note that, at submission, the dataset's temporal coverage does not live up to the aspiration of "Chronicling Germany."

Reviewers also suggest that the authors include additional comparison of the performance of their pipeline against the current state of the art.

Reviewers raise a number of smaller points which largely appear to be misunderstandings on the part of the reviewer (cf ZtnH).

Author rebuttals address a number of reviewer concerns. In particular, as discussion closed, authors had introduced 88 pages (a 15% increase) supplementing coverage over a period from 1785 to 1924 and include additional results comparing their pipeline to the state of the art. This and other improvements caused reviewers to raise their scores. Not all updates were noted in the system, and reviewers were stingy with explanations for why they were not appeased by the authors' focused, reasoned rebuttals. Your AC must exercise discretion to fill the gap. To wit:

The reviewer scores and comments underplay the many fine qualities of this paper. Complaints about dataset size expressed in pages disregard how much data is represented in each page of densely typeset text.

More importantly, the size and temporal coverage of the dataset matters because it implicates the ability of systems trained on the dataset to generalize to out-of-sample observations. The original manuscript addresses this (p. 8); a text pipeline trained on this dataset performs well on out-of-sample observations from many years before and many years after the bulk of the training data. Authors additionally compare their pipeline to the state of the art.

Authors should attend to the following guidance for revisions:
* Additionally emphasize the technical aspects of the problem of OCR on German-language text from this period,
* Emphasize how this dataset facilitates both technical work on unsolved problems and applications to historical and other social-scientific work, highlighting what we can know about this period thorugh computational history/social science with this dataset that we would not without it,
* Incorporate what authors offered in response to relevant reviewer concerns (H4jz and ffkZ) appropriately either in the main text or supplemental information of the paper,
* Focus attention on the dataset, where the pipeline demonstrates the need for the dataset by highlighting how models trained primarily on Antiqua (English) text exhibit poor OCR performance on images of newspaper text from this period.

Other than that the paper is a fine example of a dataset that bridges across social science and machine learning. It addresses technical challenges and unlocks research in a specific subject matter domain at the same time. It does both at a state-of-the-art level and for this reason should be accepted.

Note from PC: This year, the track has been incredibly competitive, which meant that many good papers had to be rejected. After careful discussion with the SACs we have concluded that this paper unfortunately cannot be accepted this time. This is the final decision, which cannot be appealed. We encourage the authors to incorporate feedback from reviewers and additional results / discussion provided during the author response period in their next submission.